# Occurrence of Chlamydiae in Corvids in Northeast Italy

**DOI:** 10.3390/ani12101226

**Published:** 2022-05-10

**Authors:** Rachid Aaziz, Karine Laroucau, Federica Gobbo, Daniela Salvatore, Christiane Schnee, Calogero Terregino, Caterina Lupini, Antonietta Di Francesco

**Affiliations:** 1Bacterial Zoonoses Unit, Animal Health Laboratory, Anses, University Paris-Est, 94700 Maisons-Alfort, France; rachid.aaziz@anses.fr (R.A.); karine.laroucau@anses.fr (K.L.); 2Istituto Zooprofilattico Sperimentale delle Venezie, 35020 Legnaro, PD, Italy; fgobbo@izsvenezie.it (F.G.); cterregino@izsvenezie.it (C.T.); 3Department of Veterinary Medical Sciences, University of Bologna, 40064 Ozzano dell’Emilia, BO, Italy; daniela.salvatore2@unibo.it (D.S.); caterina.lupini@unibo.it (C.L.); 4Institute of Molecular Pathogenesis, Friedrich-Loeffler-Institut (Federal Research Institute for Animal Health), D-07743 Jena, Germany; christiane.schnee@fli.de

**Keywords:** chlamydia, corvids, HRM, Italy, MLST, PCR

## Abstract

**Simple Summary:**

The continuous improvement of next-generation sequencing techniques has led to an expansion of the number of *Chlamydia* species, as well as their host range. Recent studies performed on wild birds have detected *Chlamydia* strains with characteristics intermediate between *Chlamydia psittaci* and *Chlamydia abortus*. In this study, 12/108 corvids tested positive for *Chlamydia* by real-time PCR. Molecular characterisation at the species level was possible for eight samples, with one positive for *C. psittaci* and seven for *C. abortus.* Considering the well-known zoonotic role of *C. psittaci* and that a potential zoonotic role of avian *C. abortus* strains cannot be excluded, people who may have professional or other contact with wild birds should take appropriate preventive measures.

**Abstract:**

*Chlamydiaceae* occurrence has been largely evaluated in wildlife, showing that wild birds are efficient reservoirs for avian chlamydiosis. In this study, DNA extracted from cloacal swabs of 108 corvids from Northeast Italy was screened for *Chlamydiaceae* by 23S real-time (rt)PCR. The positive samples were characterised by specific rtPCRs for *Chlamydia psittaci*, *Chlamydia abortus*, *Chlamydia gallinacea*, *Chlamydia avium*, *Chlamydia pecorum* and *Chlamydia suis.* Cloacal shedding of *Chlamydiaceae* was detected in 12 out of 108 (11.1%, 5.9%–18.6% 95% CI) corvids sampled. Molecular characterisation at the species level was possible in 8/12 samples, showing *C. psittaci* positivity in only one sample from a hooded crow and *C. abortus* positivity in seven samples, two from Eurasian magpies and five from hooded crows. Genotyping of the *C. psittaci*-positive sample was undertaken via PCR/high-resolution melting, clustering it in group III_pigeon, corresponding to the B genotype based on former *ompA* analysis. For *C. abortus* genotyping, multilocus sequence typing was successfully performed on the two samples with high DNA load from Eurasian magpies, highlighting 100% identity with the recently reported Polish avian *C. abortus* genotype 1V strain 15-58d44. To confirm the intermediate characteristics between *C. psittaci* and *C. abortus*, both samples, as well as two samples from hooded crows, showed the chlamydial plasmid inherent in most *C. psittaci* and avian *C. abortus*, but not in ruminant *C. abortus* strains. The plasmid sequences were highly similar (≥99%) to those of the Polish avian *C. abortus* genotype 1V strain 15-58d44. To our knowledge, this is the first report of avian *C. abortus* strains in Italy, specifically genotype 1V, confirming that they are actively circulating in corvids in the Italian region tested.

## 1. Introduction

Chlamydiae (order *Chlamydiales*, family *Chlamydiaceae*, genus *Chlamydia)* are Gram-negative obligate intracellular bacteria detected worldwide in a broad host range, including humans and livestock, as well as companion, wild and exotic animals [1].

In recent years, the continuous improvement of next-generation sequencing techniques has led to an expansion of the number of *Chlamydia* species, as well as their host range [2]. Today, the genus *Chlamydia* includes 14 species, namely *C. trachomatis*, *C. pneumoniae*, *C. psittaci*, *C. abortus*, *C*. *pecorum*, *C. suis*, *C. felis*, *C. caviae*, *C. muridarum*, *C. gallinacea*, *C. avium*, *C. serpentis*, *C. poikilothermis* and *C. buteonis* [3,4,5], plus a further four *Candidatus* (*Ca*) species, namely *Ca* C. ibidis, *Ca* C. sanzinia, *Ca* C. corallus and *Ca* C. testudinis [6,7,8,9]. In addition, the genus *Chlamydiifrater (Cf.),* with the two species *Cf. phoenicopteri* and *Cf. volucris,* was recently introduced [10].

In birds, *C. psittaci* is the longest known aetiological agent of chlamydiosis, detected in poultry, pet and free-living birds [11]. Depending on the virulence of the strain and the avian host, chlamydiosis can be subclinical or characterised by ocular, respiratory and enteric signs, with intermittent bacterial excretion, especially in stressful situations (migration, breeding, illness). The detection and differentiation of *C. psittaci* strains were initially performed by monoclonal antibody typing [12,13,14], obtaining six avian serovars (A–F). Later, the serotyping method was replaced by faster genotyping techniques, obtaining A–F genotypes and an additional genotype E/B [15]. The transition to DNA-based typing methods was facilitated by the equivalence detected in most cases between serotypes and genotypes [14]. All genotypes were considered to be readily transmissible to humans. In addition, other avian *C. psittaci* genotypes, designated 1V, 6N, Mat116, R54, YP84, CPX0308, I, J, G1 and G2, have been proposed [16,17,18]. Recently, *C. psittaci* genotype M56, originally isolated from muskrat, has also been highlighted in wild raptors [19,20].

For a long time, *C. psittaci* has been considered the only agent of chlamydiosis in birds. However, recent studies have proposed three new avian species and one *Candidatus* species: *C. gallinacea* from poultry, *C. avium* from pigeons and psittacine birds [21], *C. buteonis* from raptors [5] and *Ca* C. ibidis from African Sacred Ibis [6]. Moreover, other chlamydial agents could be involved in avian chlamydiosis, considering that *C. abortus*, *C. pecorum*, *C. trachomatis*, *C*. *suis*, *C. pneumoniae* and *C. muridarum* were molecularly detected in birds [22,23,24,25,26].

In light of this evidence and the multiple PCR-based detection methods recently developed, the purpose of this study was to investigate the presence of *Chlamydiaceae* species in corvids in Italy, and to then characterise them by fast and high discriminant molecular techniques such as species-specific real-time (rt) PCR assays, multilocus sequence typing (MLST) and PCR/high-resolution melting (HRM) analysis.

## 2. Materials and Methods

### 2.1. Sampling

From April to June 2021, 108 dead birds, including 52 Eurasian magpies (*Pica pica*), 38 hooded crows (*Corvus cornix*) and 18 Eurasian jays (*Garrulus glandarius*), from the Veneto region (northern Italy) were submitted to the Istituto Zooprofilattico Sperimentale delle Venezie for disease surveillance activities (such as Avian Influenza Virus, West Nile Virus and Usutu virus). A cloacal swab was collected from each bird carcass and immediately stored at −20 °C.

### 2.2. DNA Extraction

Total DNA was individually extracted from each sample using the QIAamp DNA mini kit (Qiagen, 40724 Hilden, Germany), following the manufacturer’s instructions. Positive (*C. psittaci* Loth strain) and negative (kit reagents only) extraction controls were included in each set of extraction. The DNA extracts were stored at −20 °C before analysis.

### 2.3. Real-Time PCRs

All DNA extracts were screened using a *Chlamydiaceae*-specific rtPCR targeting the 23S rRNA gene fragment [27]. An analytical cut-off value was selected at a cycle threshold (Ct) of 39.

All samples that gave a positive signal with the 23S-rtPCR were re-examined with in-house-specific *enoA*-based *C. psittaci* and *enoA*-based *C. abortus* rtPCRs, developed and already in use in Anses laboratory in order to improve the detection of *C. psittaci* and avian *C. abortus* strains compared to traditional methods. For the specific *C. psittaci* detection, we used *enoA*_CpsF43 5′-ATTCGCCCTATAGGTGCACAT-3′ and *enoA*_CpsR162 5′-GCCTTCATCTCCAACTCCTGTAG-3′ primers and the probe *enoA*_CpsP79 5′-[FAM] GTGCGTATGGGTGCTGATGTTT [BHQ1]-3′. For the specific detection of *C. abortus*, we used *enoA*_CabF13 5′-AACAACGGCCTGCAATTTCAAG-3′and *enoA*_CabR124 5′-TGAGAAGGTTTTTCAATGTATGGAAC-3′ primers, as well as the probe *enoA*_CabP93 5′-[FAM] GGCACCCATACGTACAGCTTCTTG [BHQ1]-3′. DNA amplification was performed in a final volume of 20 µL containing 10 µL of TaqMan^TM^ Fast Advanced Master Mix (Applied Biosystems, Waltham, MA, USA), 0.6 µM of each primer, 0.1 µM of the probe, 2 µL of DNA sample and water (qsp 20 µL). The reaction of rtPCR was carried out in 7500 or ViiA7 apparatus (Applied Biosystems, Waltham, MA, USA) using the following cycling parameters: 50 °C for 2 min, 95 °C for 20 s, 45 cycles of 95 °C for 3 s and 60 °C for 30 s. Specificity and sensitivity of in-house *enoA*-based *C. psittaci* and *enoA*-based *C. abortus* rtPCRs are shown in Appendix A and Appendix A, respectively.

The 23S rRNA-positive samples were also tested with an *enoA*-based *C. gallinacea*-specific rtPCR according to Laroucau et al. [28], a 16S rRNA-based *C. avium* rtPCR [29], a 23S rRNA-based *C. suis* and an *ompA*-based *C. pecorum* rtPCRs [30].

### 2.4. Multilocus Sequence Typing (MLST) Analysis

The MLST analysis was performed on the *C. abortus*-positive samples, according to Pannekoek et al. [31,32]. Fragments of seven housekeeping genes, namely *gatA*, *oppA*3, *hflX*, *gidA*, *enoA*, *hemN* and *fumC,* were amplified and sequenced using the primers and conditions described on the Chlamydiales MLST website [33]. Sanger sequencing of both DNA strands was performed by Eurofins Genomics (85560 Ebersberg, Germany), and the numbers for alleles and the sequence type (ST) were assigned in accordance with the *Chlamydiales* MLST database and uploaded on the PubMLST website [33]. For each sample, DNA sequences of the seven alleles were manually assembled to obtain 3098 nucleotides. Multiple alignments of the seven concatenated MLST gene fragments with a large panel of *C. psittaci* and *C. abortus* strains were performed using the MEGA7 software [34]. Phylogenetic trees were constructed by the maximum likelihood method based on the general time-reversible model [34].

### 2.5. Detection of the Chlamydial Plasmid

The presence of the plasmid was investigated on the *C. abortus*-positive samples by a conventional PCR, using the in-house primer set (pCpsi_Fw 5′-AGCTGTGCATACATGGCTGT-3′ and pCpsi_Rv 5′-CAGTAACTGCGGTAGCTCGT-3′), targeting a 734-nucleotide region within the chlamydial plasmid tyrosine recombinase *XerC* gene harboured by the plasmid II of *C. abortus* strain 15-58d44 (GenBank Accession Number OU508368.1). DNA amplification was performed in a final volume of 25 µL containing 2 µL of DNA samples, 1× PCR reaction buffer, 1 U of Hot start *Taq* DNA polymerase (Qiagen, 40724 Hilden, Germany), 200 µM of each deoxynucleotide triphosphate (Promega, 20126 Milan, Italy) and 0.4 mM of each forward and reverse primers. The following cycling parameters were used: initial denaturation at 94 °C for 10 min, 40 cycles of 94 °C for 30 s, 50 °C for 30 s, 72 °C for 60 s, final extension at 72 °C for 7 min. DNA amplified fragments were sequenced by the Sanger method by Eurofins Genomics (85560 Ebersgerg, Germany) using the same primers. Nucleotide sequences of the plasmid DNA fragments were aligned and analysed in MEGA7 [34]. Phylogenetic trees were constructed by using the maximum likelihood method based on the general time-reversible model. Bootstrap tests were for 1000 repetitions.

### 2.6. PCR/High-Resolution Melting (HRM) Analysis

Genotyping of the *C. psittaci*-positive samples was undertaken via PCR/high-resolution melting (HRM), performed according to Vorimore et al. [35].

## 3. Results

### 3.1. Results of the 23S rtPCR

The results are shown in Table 1. The PCR targeting the 23S rRNA gene fragment showed cloacal shedding of *Chlamydiaceae* in 12 out of 108 (11.1%, 5.9%–18.6% 95% CI) birds sampled. The *Chlamydiaceae* prevalence was higher among hooded crows (9/38, 23.7%, 11.4%–40.2% 95% CI) than Eurasian magpies (2/52, 3.8%, 0.5%–13.2% 95% CI) and Eurasian jays (1/18, 5.6%, 0.1%–27.3% 95% CI). A mean Ct value of 35.1 (Ct range from 26.1 to 38.5) was observed.

### 3.2. Molecular Characterisation of Chlamydiaceae-Positive Samples

Eight out of twelve *Chlamydiaceae*-positive samples were characterised by the species-specific rtPCRs. Only one DNA sample from a hooded crow was positive for *C. psittaci* rtPCR. Interestingly, *C. abortus* DNA was detected in 5 of 9 (55.5%) and 2/2 (100%) *Chlamydiaceae*-positive hooded crow and Eurasian magpie samples, respectively.

No positive results were shown for *C. avium*, *C. gallinacea*, *C. pecorum* or *C. suis*.

With respect to the remaining four *Chlamydiaceae*-positive samples, the three samples from hooded crows showed a signal above the cut-off value of 39 when tested with rtPCR for *C. abortus,* while the sample from Eurasian jays showed no signal to specific rtPCRs.

Further attempts at characterisation of these samples by *Chlamydiales* 16S rRNA PCR or 23S rRNA PCR were unsuccessful, possibly due to the low amount of DNA.

### 3.3. Genotyping of C. abortus-Positive Samples by MLST and Plasmid Sequencing

The MLST was successfully performed only on the two *C. abortus*-positive DNA samples from Eurasian magpies that had shown high levels of chlamydial excretion by rtPCR. The MLST sequences obtained from the two Eurasian magpie samples were identical and identified as ST152. Regarding the five hooded crow samples positive for *C. abortus*, for three of them, it was possible to amplify only three gene sequences (*gidA*, *enoA*, *hemN*) that were identical to those of the two Eurasian magpie samples. Regarding the remaining two hooded crow samples positive for *C. abortus*, the low quantity of DNA did not allow an appreciable MLST result. Comparative phylogenetic analysis of the concatenated MLST sequences of the two Eurasian magpie samples and a large panel of *C. psittaci* and *C. abortus* strains, including avian and ruminant *C. abortus* strains, showed a topology Identical to that of the avian *C. abortus* genotype 1V strain 15-58d44 recently detected in Poland [18] (Figure 1). MLST sequences were uploaded to the PubMLST database, and their ST allelic profile can be consulted [33]. The sequencing of the plasmid DNA fragment of the *XerC* gene was successfully performed for the two Eurasian magpie samples and two hooded crow samples, showing 100% identity (Eurasian magpie samples) and 99.9% similarity (hooded crow samples) with the same fragment of the 15-58d44 plasmid (Figure 2). The plasmid sequences obtained in this study were submitted to the GenBank database and are available under the following accession numbers: ON165250-ON165253.

### 3.4. Chlamydia psittaci Genotype Identification

The PCR/high-resolution melting performed on the *C. psittaci*-positive sample obtained from one hooded crow was consistent to the group III_pigeon, corresponding to the B genotype based on former *omp*A analysis.

## 4. Discussion

In Europe, recent decades have seen an adaptation of wild animal populations to specific conditions of the urban environment [36]. Corvids are resident or short-range migratory birds characterised by a generalist behaviour less demanding regarding the environment and the feed [37]. Thus, they have developed a marked synanthropic temperament, taking advantage of the presence of crops, waste and landfills, all of which are derived from human activities. These features are the basis of their successful adaptation to urban ecological niches and their increase, in contrast to the decrease in many bird species due to pollution or habitat modification/destruction. In Italy, for years, there has been an increase in the population levels of corvids, both in urban and rural environments. In the period 2000–2020, an average annual variation of 0.80% (±0.12) for hooded crows and 2.05% (±0.13) for Eurasian magpies has been registered. The conservation status was considered favourable for both avian species [38].

The *Chlamydiaceae* occurrence in the Corvidae family has been investigated in recent studies [18,20,24,39], showing prevalence values ranging from 5.1% to 29%.

In Italy, a previous investigation [24] performed on 76 corvids showed that 22/76 (29%) birds were PCR *Chlamydia* positive, with only one *C. psittaci*-positive sample vs. 21 *C. suis*-positive animals. The source of *C. suis* had been related to contact with wild boar, since the corvids sampled were from hilly areas where the presence of wild boar was consistently reported. Some samples showed mixed sequences of *C. psittaci-C. abortus* but were not included in the study due to the low amount of DNA, impeding further investigations (data not shown).

The results of the present study, performed in another geographical area, confirm *Chlamydiaceae* circulation in corvids. *Chlamydia psittaci* positivity was shown in only one DNA sample from a hooded crow, according to the low *C. psitttaci* prevalence previously detected [24]. Interestingly, the presence of avian *C. abortus* strains was detected in most of the *Chlamydiaceae*-positive samples (7/8, 87.5%) characterised at the species level. In addition, three *Chlamydiaceae*-positive samples from hooded crows reacted to *C. abortus* rtPCR, but they were not included in the prevalence calculation due to the higher signal cut-off value. The rtPCR results were confirmed by MLST analysis on the two strongest positive samples, allowing to group the two samples with avian *C. abortus* genotype 1V strain 15-58d44 [18]. To confirm the intermediate characteristics between *C. psittaci* and *C. abortus*, in four *C. abortus*-positive samples, it was possible to partly sequence the chlamydial plasmid inherent in most *C. psittaci* and avian *C. abortus*, but not in ruminant *C. abortus* strains. The fragment plasmid sequences were closely related (≥99%) to those of the Polish avian *C. abortus* genotype 1V strain 15-58d44 [18]. To our knowledge, this is the first report of avian *C. abortus* strains in Italy, specifically genotype 1V, which appear to be actively circulating in wild bird populations of the Veneto region, at least in the avian species considered in this study. So far, avian *C. abortus* strains have been reported in mallard (*Anas platyrhynchos*), swan (*Cignus*) and Eurasian teal (*Anas crecca*), as well as in Eurasian magpie and hooded crow, in Poland [18]; in rook (*Corvus frugilegus*) and Korean magpie (*Pica sericea*) in South Korea [39]; and in common buzzard (*Buteo buteo*) and carrion crow (*Corvus corone*), as well as in rook, in Switzerland [20]. Furthermore, some *C. psittaci* isolates from parrots and parakeets were shown to differ from classical avian *C. psittaci* strains and to be more closely related to *C. abortus* species [32,40]. Recently, the sequencing of the whole genome of one of these isolates allowed exploring its evolutionary relationship to both *C. psittaci* and *C. abortus*, supporting its reclassification as *C. abortus* species [41]. These acquisitions, if supported by further studies using next-generation sequencing, could suggest some changes in the taxonomy of the *Chlamydiaceae* family [2]. Considering that data on the prevalence of avian *C. abortus* strains, as well as their host and geographical distributions, are still limited, our results could supplement the current literature.

At the present state of knowledge, the zoonotic impact of the avian *C. abortus* strains has not been investigated; however, it cannot be excluded, considering their phylogenetic relationship with *C. psittaci* and ruminant *C. abortus* strains. The zoonotic role of *C. psittaci* has been known for a long time. Zoonotic transmission occurs by the inhalation of respiratory secretions or dried faeces dispersed in the air as fine droplets or dust particles, as well as through handling infected birds [42], particularly in high-risk individuals, such as veterinarians, bird breeders and pet-shop or poultry workers. In humans, symptoms range from mild illness to atypical pneumonia or to serious complications in internal organs [43]. *Chlamydia abortus* has recently been detected in a wide range of animals, so far mainly associated with enzootic abortions in small ruminants [44]. Most reported cases of human *C. abortus* infection involve pregnant women, initially showing an influenza-like illness with the consecutive development of placental dysfunction, leading to foetal death, as a result of direct or indirect contact with infected animals [45,46,47,48]. Extra-gestational infections of *C. abortus* manifested as pelvic inflammatory disease have also been described [49]. More recently, *C. abortus* was suggested to be the probable causative agent of atypical pneumonia detected in a veterinary researcher working in a laboratory where experimental intranasal infections with *C. abortus* were developed in sheep [50]. In conclusion, the circulation of a well-known zoonotic agent as *C. psittaci* and other potential zoonotic chlamydiae in corvid populations should be considered by workers in wildlife centres and people who may have professional or other contact with wild birds, who are urged to take appropriate preventive measures [51].

## 5. Conclusions

The results of this study on corvids confirm the *Chlamydiaceae* circulation previously detected in another Italian area [24]. Interestingly, *C. psittaci* was detected in only one sample, whereas most of the positive samples showed high molecular similarity with avian *C. abortus* strains, specifically genotype 1V. Further investigations involving more Italian geographical areas and more wild avian species are needed, to confirm and deepen these results.

Moreover, considering the constant increase of the populations of corvids in urban/peri-urban areas and the consequent possibility of human professional and non-professional contacts, the potential zoonotic role of the avian *C. abortus* strains must be further investigated.

## Figures and Tables

**Figure 1 animals-12-01226-f001:**
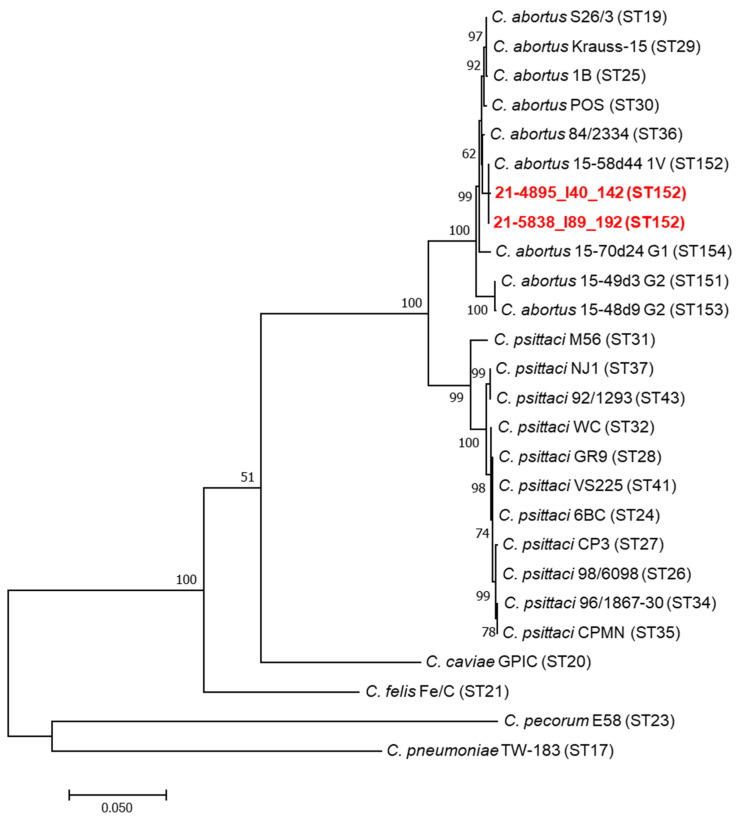
Phylogenetic analyses of multilocus sequence typing (MLST) concatenated sequences of Chlamydia. Concatenated sequences (3098 nucleotides) were aligned and analysed in MEGA7. Phylogenetic trees were constructed by using the maximum likelihood method based on the general time-reversible model. Bootstrap tests were for 1000 repetitions. Numbers on tree nodes indicate bootstrap values of the main branches. Horizontal line scale indicates the number of nucleotide substitutions per site. The MLST sequence type (ST) is indicated. The red colour represents the avian *C. abortus* strains analysed in this study.

**Figure 2 animals-12-01226-f002:**
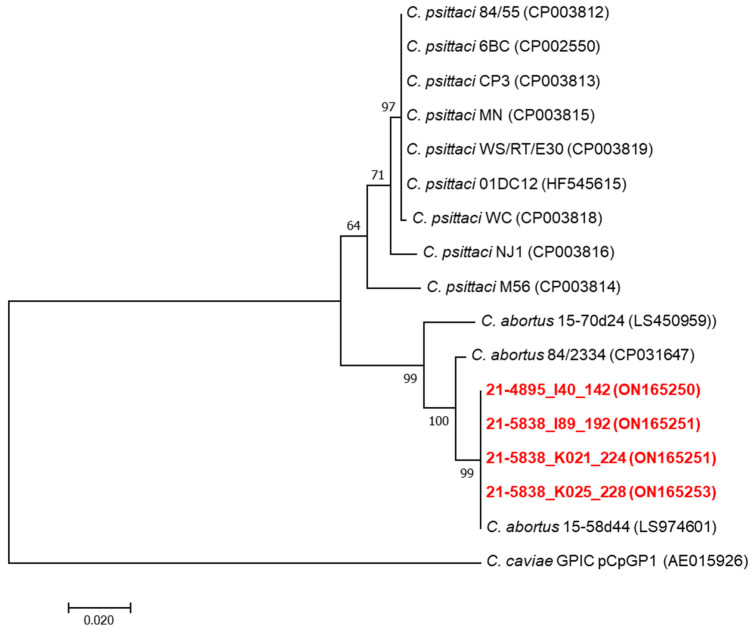
Phylogenetic analyses of plasmid *XerC* gene fragment sequences of Chlamydia. Nucleotide sequences (734 nucleotides) were aligned and analysed in MEGA7. Phylogenetic trees were constructed by using the maximum likelihood method based on the general time-reversible model. Bootstrap tests were for 1000 repetitions. Numbers on tree nodes indicate bootstrap values of the main branches. Horizontal line scale indicates the number of nucleotide substitutions per site. The outgroup is represented by *C. caviae* plasmid DNA fragment. Accession numbers of different *Chlamydia* are indicated. The red colour represents the avian *C. abortus* strains analysed in this study.

**Table 1 animals-12-01226-t001:** Total number and percentage of *Chlamydiaceae*-positive corvids per species and number and percentage of chlamydial species identified.

Corvid Species	Samples	23S rtPCR	*C. psittaci* rtPCR	*C. abortus* rtPCR	*C. gallinacea* rtPCR	*C. avium* rtPCR	*C. pecorum* rtPCR	*C. suis* rtPCR	Non-Classified *Chlamydia*
Eurasian magpie	52	2 (3.8%)	-	2 (3.8%)	-	-	-	-	
Hooded crow	38	9 (23.7%)	1 (2.6%)	5 (13.1%)	-	-	-	-	3 (7.9%)
Eurasian jays	18	1 (5.6%)	-	-	-	-	-	-	1 (5.6%)
Total	108	12 (11.1%)	1 (0.9%)	7 (6.5%)	-	-	-	-	4 (3.7%)

## Data Availability

The plasmid sequences generated in this study are available in GenBank under Accession Numbers ON165250-ON165253; ST allelic profile of MLST sequences can be consulted on the PubMLST database.

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
