# Peer review of "Occurrence of Chlamydiae in Corvids in Northeast Italy"

_animals, 2022, doi:10.3390/ani12101226_

Round 1
Reviewer 1 Report
Dear authors, this is a good reaserch work that is highly seeked for publication because of its importance. I would like to congratulate with you, despite the simplicity of the study, for the high quality manuscript you have submitted to Animals (MDPI). This is in line with other manuscripts received from Istituto Zooprofilattico.
Here attached you can find a file with the required revisions. Best, the reviewer

Author Response
Dear Sir/Madame,
thank you for your suggestions that helped us improve the paper.
The manuscript has been revised as follows:
General: add “Eurasian” before magpies throughout the ms. For example lines 31, 33, 90… Maintain the full common name in the abstract and at line 90. Then, if you want, you can write (hereafter, magpies) and just abbreviate it.
The text has been modified as suggested
Lines 32 and 39: no need for abbreviation in the abstract. Remove MLST and HRM. Thanks
The text has been modified
Scientific names at lines 52, 78 and so on should be put in italics. Thanks
“The word ‘Candidatus’, but not the genus name and/or the vernacular epithet, is printed in italics. ‘Candidatus’ is often abbreviated ‘Ca’” (Oren A. Nomenclature of prokaryotic ‘Candidatus’ taxa: establishing order in the current chaos. New Microbes and New Infections 2021, 44, 100932)
Line 91: As above, Eurasian jays.
The text has been modified
Line 109: Missing reference. Please add
The in house specific enoA-based Chlamydia psittaci and enoA-based Chlamydia abortus qPCRs have been developed in Anses laboratories and are first published in this paper, so there are no references to mention
Lines 110-115, 136-137,…: Remove the spaces between the nucleotides: no need to separate “codons”.
The text has been modified
Lines 115-116: “Standard 115 cycling conditions were applied.” Does not fit with methods section. Just provide all the details, such as the number of amplification cycles, Ta, T of elongation, denaturation step, final elongation step as well as with DNA polymerase/master mix have you used with the company name between brackets.
All the details have been added
Line 127: Specify the type of sequencing used. I suppose it is Sanger sequencing. In that case, just start the phrase as “Sanger sequencing was…”.
The type of sequencing was indicated
Line 128: Put the link between brackets. Thanks
The text has been modified
Line 129: Provide more information about the trimming process.
The numbers for alleles and the sequence type (ST) were assigned in accordance with the Chlamydiales MLST Database where they were uploaded on. For each sample, DNA sequences of the seven alleles were manually assembled to obtain 3,098 nucleotides. Multiple alignments of the seven concatenated MLST gene fragments with a large panel of C. psittaci and C. abortus strains were performed using the MEGA7 software [34]. Phylogenetic trees were constructed by the Maximum Likelihood method based on the General Time Reversible model [34]. The information were added in the text.
Line 131: Check how to cite Mega7 tool and cite it. Thanks
The wording has been corrected. MEGA7 has been cited in the reference n. 34
Line 138: Specify which gene, specify all the setting of the PCR reaction. Again specify sequencing method. Citation missing (see https://blast.ncbi.nlm.nih.gov/Blast.cgi?CMD=Web&PAGE_TYPE=BlastDocs&DOC_TYPE=References). Thanks
Gene type, PCR reaction and sequencing method were added
Line 158: Check for typos/spelling “low a signal”
The text has been modified
Line 180: Substitute the comma with a full stop. 0,050 → 0.050
The comma has been replaced
Line 196: It is good to abbreviate scientific names but not when a phrase start with them. C. → Chlamydia
The text has been modified
Line 202: What does “an adjustment” mean in this context? Please be the sentence clearer. Thanks
The text has been modified
Line 204: Sorry, are you writing in British or American English? You wrote color (that is US, colour -> UK), but behaviour (that is UK). Please, do not a mix of the two styles. Adjust as you want. Thanks, and sorry for being so finicky. Pay attention also to foetal at line 264 and centres at line 272 depending on what style you will choose.
The text has been modified
Lines 205, 207, 209, 210: Missing citations.
Two references have been added
Line 214: The family name of animals do not need italics. This is only for viruses (e.g., Corvidae; Flaviviridae). Remove italics.
The text has been modified
Line 224: after mentioning common names put, between brackets, the scientific names in italics. Same for line 235. Please note that the plural of boar (cinghiale) do no exist in English. So one wild boar, 10 wild board. Just correct accordingly (remove the s) in lines 234, 235…
The text has been modified
Line 252: Since they are limited just put some of them as citation at the end of this sentence. Thanks
The text has been modified
Line 256: Missing citation
Citation n. 43
Line 263: Influenza-like. Add the hyphen
The text has been modified

Reviewer 2 Report
This study by Aaziz and co-authors describes a study investigating the prevalence of Chlamydiae in corvid birds in the Northeast of Italy. This elegant short paper describes use of high resolution molecular techniques to identify Chlamydiae positive samples and then to speciate them with MLST and phylogenetic analysis of genome and plasmid sequences. Some of the positive samples have been attributed to the emerging intermediate avian C. abortus strains associating with other published strains with distinct molecular sequence differences to both C. psittaci and C. abortus. There are a few typographical suggestions to be addressed as follows:
Line 82: Amend the sentence to read "In light of this evidence, and the..."
Line 84: Amend the sentence to read ..." species in corvids in Italy; and to then characterise them by fast and high....."
Line 104: State that the "cut-off value" is a 'cycle threshold value' in the sentence.
Table 1:
Title "Non-classified Chlamydia" keep whole words on a new line for presentation.
Line 158: Amend to read "jays, showed a signal level beneath the cut-off threshold."
Figure 1, line 181: Write MLST in full before abbreviating it in lines 181 and 185.
Line 182: Write 'nucleotide' in full, no need to abbreviate it.
Reference number 33, please adjust website address, Chlamydiales-spp has a capital 'C'.
Author Response
Dear Sir/Madame,
thank you for the kind comments.
The text has been revised following your suggestion:
This study by Aaziz and co-authors describes a study investigating the prevalence of Chlamydiae in corvid birds in the Northeast of Italy. This elegant short paper describes use of high resolution molecular techniques to identify Chlamydiae positive samples and then to speciate them with MLST and phylogenetic analysis of genome and plasmid sequences. Some of the positive samples have been attributed to the emerging intermediate avian C. abortus strains associating with other published strains with distinct molecular sequence differences to both C. psittaci and C. abortus. There are a few typographical suggestions to be addressed as follows:
Line 82: Amend the sentence to read "In light of this evidence, and the..."
The text has been modified
Line 84: Amend the sentence to read ..." species in corvids in Italy; and to then characterise them by fast and high....."
The text has been modified
Line 104: State that the "cut-off value" is a 'cycle threshold value' in the sentence.
The text has been modified
Table 1:
Title "Non-classified Chlamydia" keep whole words on a new line for presentation.
The table has been modified
Line 158: Amend to read "jays, showed a signal level beneath the cut-off threshold."
The text has been modified
Figure 1, line 181: Write MLST in full before abbreviating it in lines 181 and 185.
The text has been modified
Line 182: Write 'nucleotide' in full, no need to abbreviate it.
Nucleotide has been modified in full
Reference number 33, please adjust website address, Chlamydiales-spp has a capital 'C'.
Reference has been corrected
Reviewer 3 Report
This is interesting report expanding on avian C abortus strains detected in corvids from Italy, where the infecting strains are same ST as those previosly described. I do have suggestions and comments for the authors in order to improve clarity.
Line 59: you could streamline the introduction bit more by reducing Cps ompA genotyping (i.e Cps ompA genotyping confirms the genetically diverse genotypes), and by placing focus on avian C abortus strains (their genetic diversity, hosts). Also has Cps been detected in corvids yet?
Line 90: Are ethics (swab collection approval) needed for sampling dead birds? Or is exempted?
Line 103: maybe not need to have rt abbreviation as not to confuse with reverse transcription? maybe qPCR?
Line 128: Are these deposited in PubMLST to be publicly available? Or Genbank? Accession number is needed when reporting sequence data.
Line 131:In the figure legend is says phylogenetic tree. Looks to me as a tree with node support. But here is says dendrogram (which is slightly different that phylo tree in this context). Please correct
Line 137: which plasmid gene ? CDS1 to 8? this information needs to be added. here add phylogenetic methods descriptions (as plasmid fragments were used in phylogenetic analyses below) I get it is in the figure legend but must be outlined here too for clarity.
Line 143: To determine C. psittaci genotype, HRM analysis was used, which is okay, but from my understanding, the ability to differentiate between SNPs this way differs by 0.3-1oC (Vorimore et al. 2021). Would be preferable to at least attempt to amplify and sequence the whole or partial ompA gene for their C. psittaci pos sample to be 100% sure and confirm genotype that way. (As studies show diversity within B type genotypes)
Line 147: When discussing prevalence and detection levels of Chlamydiaceae in the birds, you should provide confidence intervals for prevalence, as well as report true prevalence and apparent prevalence in a table nominated test sensitivity and specificity.
Line 158: The 4 unresolved Chlamydiaceae-positive samples, 3 from a hooded crow and 1 from a Eurasian Jay - it would be interesting to attempted to see what these could be, by Chlamydiales 16S rRNA PCR (short or long fragment) or via 23S rRNA PCR - just to get some sense of identity on these unresolved samples.
Line 160: The is a really low likelihood of C suis in birds (cloaca), unless some rare spillover event. Curious why you screened in the first place?
Lines 174- 175: its a similar to that fragment not the whole plasmid, be exact. its a plasmid fragment seq (which plasmid CDS?) Add this information in M and M
Line 183: Horizontal scale indicates what measure of genetic distance (I believe its the number of nucleotide substitutions per site in your analyses as you are using ML but i could be wrong). Should be exact.
Line 189: not a plasmid, but rather 700nt fragment of plasmid CDSX . Please correct accordingly. Also denote your outgroup.
Line 198: Would be good to supplement with ompA seq.
Disscussion: could be bit more streamlined to in context of your findings. It more revises the previous literature rather the paper.
Line 207: A reference to support this? or for further interest of the reader?
Line 216: Syntax of this sentence: revise to: Chlam_POS samples from corvids
Line 235: Ahh, got it, this answers my comment above. interesting interface for spillover.
Line 245-246: Revise these sentences, it was 700nt fragment of plasmid CDSX not the whole plasmid, interpretation could be misleading. This gives you clue that plasmid is conserved and these are plasmid-bearing strains. .
Also syntax of this sentence: just say chlamydial plasmid (as it is so well characterized)
Line 249: Again, bit over-interpretative. It was 3 bird species you assessed so OK for these but maybe to broad to extrapolate for wild birds in the region.
Line 276: Again, revise this statement considering you sample number and detection level. yes you confirm that avian Cab infect free-range corvids from Italy but not sure is whole population is infected. This could yours future direction to perform Italy wide surveillance in free-range corvids.
Author Response
Dear Sir/Madame,
Thank you for your suggestions which helped us improve the paper.
The manuscript has been modified as follows:
Line 59: you could streamline the introduction bit more by reducing Cps ompA genotyping (i.e Cps ompA genotyping confirms the genetically diverse genotypes), and by placing focus on avian C abortus strains (their genetic diversity, hosts). Also has Cps been detected in corvids yet?
The text has been modified.
Line 90: Are ethics (swab collection approval) needed for sampling dead birds? Or is exempted?
The opinion of the Ethics Committee is not necessary if investigations are carried out on dead animals.
Line 103: maybe not need to have rt abbreviation as not to confuse with reverse transcription? maybe qPCR?
We would prefer to keep ”rt” since it stands for real time and should not be mixed up with RT= reverse transcription; qPCR instead stands for quantitative PCR and is reserved to real quantitative analysis such as expression analysis by PCR.
Line 128: Are these deposited in PubMLST to be publicly available? Or Genbank? Accession number is needed when reporting sequence data.
The MLST sequences have been uploaded on the PubMLST database as required and mentioned in the text.
Line 131: In the figure legend is says phylogenetic tree. Looks to me as a tree with node support. But here is says dendrogram (which is slightly different that phylo tree in this context). Please correct
The text has been corrected.
Line 137: which plasmid gene? CDS1 to 8? this information needs to be added. here add phylogenetic methods descriptions (as plasmid fragments were used in phylogenetic analyses below) I get it is in the figure legend but must be outlined here too for clarity.
The PCR system for the amplification of the avian C. abortus plasmid DNA fragment targets the Tyrosine recombinase XerC gene harboured by the plasmid II of C. abortus strain 15-58d44 (Genebank accession number OU508368.1). This has been added in the text.
“Nucleotide sequences of the plasmid DNA fragments were aligned and analysed in MEGA7. Phylogenetic trees were constructed by using the Maximum Likelihood method based on the General Time Reversible model. Bootstrap tests were for 1000 repetitions” was added in the text.
Line 143: To determine C. psittaci genotype, HRM analysis was used, which is okay, but from my understanding, the ability to differentiate between SNPs this way differs by 0.3-1oC (Vorimore et al. 2021). Would be preferable to at least attempt to amplify and sequence the whole or partial ompA gene for their C. psittaci pos sample to be 100% sure and confirm genotype that way. (As studies show diversity within B type genotypes)
Typing of C. psittaci strains was initially based on ompA gene sequencing, but access to genomic sequences has since allowed alternative typing methods to be proposed, including MLST typing (2010) and more recently PCR-HRM typing. Indeed, this last method is based on the search for specific SNPs of the main groups identified in birds, by a simple PCR analysis and an analysis of the dissociation temperature of the PCR fragments. As expected, the typing result obtained in our study with the PCR-HRM method, on a sample from one hooded crow, is consistent with the avian host species. In our paper describing the new PCR-HRM method for C. psittaci genotyping, several pigeon strains were analysed, with a strict concordance between ompA and PCR-HRM clustering. The distinction between E or B ompA-base genotypes, both included in PCR-HRM group III_pigeon, is not essential as the host of origin matters. One of the advantages of PCR-HRM is to be able to type on biological samples even if they are not very concentrated (amplification of a fragment of about 100 bp), whereas for a fine and reliable analysis the ompA sequence (more than 1000 bp) is necessary. The bacterial load of the considered sample does not allow to consider an efficient amplification of a such long fragment (confirming the interest of the PCR-HRM).
Line 147: When discussing prevalence and detection levels of Chlamydiaceae in the birds, you should provide confidence intervals for prevalence, as well as report true prevalence and apparent prevalence in a table nominated test sensitivity and specificity.
Confidence intervals were added. Regarding to the true/apparent prevalence, not all data are available to calculate the true prevalence. The prevalence indicated is apparently the apparent prevalence.
Line 158: The 4 unresolved Chlamydiaceae-positive samples, 3 from a hooded crow and 1 from a Eurasian Jay - it would be interesting to attempted to see what these could be, by Chlamydiales 16S rRNA PCR (short or long fragment) or via 23S rRNA PCR - just to get some sense of identity on these unresolved samples.
We have now specified in the text that 3 of the 4 samples not characterized at the species level were positive for C. abortus qPCR, but with a signal above the cut-off value of 39. These samples are indeed of interest, but unfortunately, due to a low DNA content, 16S rRNA and 23S rRNA sequences (all long fragments) could not be amplified, as added in the text.
Line 160: The is a really low likelihood of C suis in birds (cloaca), unless some rare spillover event. Curious why you screened in the first place?
Chlamydia suis was investigated because of a previous survey performed in Italy showing a high percentage of positivity for C. suis in corvids.
Lines 174- 175: its a similar to that fragment not the whole plasmid, be exact. its a plasmid fragment seq (which plasmid CDS?) Add this information in M and M
The text has been modified.
Line 183: Horizontal scale indicates what measure of genetic distance (I believe its the number of nucleotide substitutions per site in your analyses as you are using ML but i could be wrong). Should be exact.
The text has been modified.
Line 189: not a plasmid, but rather 700nt fragment of plasmid CDSX . Please correct accordingly. Also denote your outgroup.
The text has been corrected as well as the outgroup has been included.
Line 198: Would be good to supplement with ompA seq.
Please see our comment above, due to a low DNA content it was not possible to amplify this long gene. The PCR-HRM typing method is a good alternative for such samples.
Discussion: could be bit more streamlined to in context of your findings. It more revises the previous literature rather the paper.
The text has been modified.
Line 207: A reference to support this? or for further interest of the reader?
Two references have been added.
Line 216: Syntax of this sentence: revise to: Chlam_POS samples from corvids
The text has been modified.
Line 235: Ahh, got it, this answers my comment above. interesting interface for spillover.
Line 245-246: Revise these sentences, it was 700nt fragment of plasmid CDSX not the whole plasmid, interpretation could be misleading. This gives you clue that plasmid is conserved and these are plasmid-bearing strains.
Also syntax of this sentence: just say chlamydial plasmid (as it is so well characterized)
The text has been modified.
Line 249: Again, bit over-interpretative. It was 3 bird species you assessed so OK for these but maybe to broad to extrapolate for wild birds in the region.
The text has been modified.
Line 276: Again, revise this statement considering you sample number and detection level. yes you confirm that avian Cab infect free-range corvids from Italy but not sure is whole population is infected. This could yours future direction to perform Italy wide surveillance in free-range corvids.
The text has been modified.
Reviewer 4 Report
The manuscript is revealing for the knowledge of Chlamydia infection in birds. However, some methodological problems need to be resolved. One of the primers could not identify the plasmid binding site because it recognizes different places. Due to the above, it is necessary to know the molecular size of the amplification fragment, so I suggest a table that reports the molecular size of the product. Furthermore, discussing the strains that were not identified both in the results and the discussion is required.

Author Response
Dear Sir/Madame,
Thank you for your suggestions.
The manuscript has been modified as follows:
The manuscript is revealing for the knowledge of Chlamydia infection in birds. However, some methodological problems need to be resolved. One of the primers could not identify the plasmid binding site because it recognizes different places. Due to the above, it is necessary to know the molecular size of the amplification fragment, so I suggest a table that reports the molecular size of the product. Furthermore, discussing the strains that were not identified both in the results and the discussion is required.
The molecular size was included (734 nucleotides).
The unidentified strains were discussed in the results and discussion
Round 2
Reviewer 3 Report
Thank you for all corrections. The manuscript is now clear and reads well. Overall this is very interesting study, and expands our knowledge on avian C abortus strains and infections.